# Improving Transformer with an Admixture of Attention Heads

**Tan M. Nguyen**[*]
Department of Mathematics
University of California, Los Angeles
tanmnguyen89@ucla.edu

**Tam Nguyen**[*]
FPT Software AI Center
nguyenminhtam9520@gmail.com

**Hai Do**
FPT Software AI Center
haidn6@fsoft.com.vn

**Khai Nguyen**
Department of Statistics and Data Sciences
University of Texas at Austin
khainb@utexas.edu

**Vishwanath Saragadam**
Department of ECE
Rice University
vishwanath.saragadam@rice.edu

**Minh Pham**
Department of Mathematics
University of California, Los Angeles
minhrose@ucla.edu

**Duy Khuong Nguyen**
FPT Software AI Center
khuongnd6@fsoft.com.vn

**Nhat Ho**[**]
Department of Statistics and Data Sciences
University of Texas at Austin
minhnhat@utexas.edu

**Stanley J. Osher**[**]
Department of Mathematics
University of California, Los Angeles
sjo@math.ucla.edu

## Abstract

Transformers with multi-head self-attention have achieved remarkable success in sequence modeling and beyond. However, they suffer from high computational and memory complexities for computing the attention matrix at each head. Recently, it has been shown that those attention matrices lie on a low-dimensional manifold and, thus, are redundant. We propose the Transformer with a Finite Admixture of Shared Heads (FiSHformers), a novel class of efficient and flexible transformers that allow the sharing of attention matrices between attention heads. At the core of FiSHformer is a novel finite admixture model of shared heads (FiSH) that samples attention matrices from a set of global attention matrices. The number of global attention matrices is much smaller than the number of local attention matrices generated. FiSHformers directly learn these global attention matrices rather than the local ones as in other transformers, thus significantly improving the computational and memory efficiency of the model. We empirically verify the advantages of the FiSHformer over the baseline transformers in a wide range of practical applications including language modeling, machine translation, and image classification. On the WikiText-103, IWSLT'14 De-En and WMT'14 En-De, FiSHformers use much fewer floating-point operations per second (FLOPs), memory, and parameters compared to the baseline transformers.

[*] Co-first authors. [**] Co-last authors. Please correspond to: tanmnguyen89@ucla.edu

36th Conference on Neural Information Processing Systems (NeurIPS 2022).

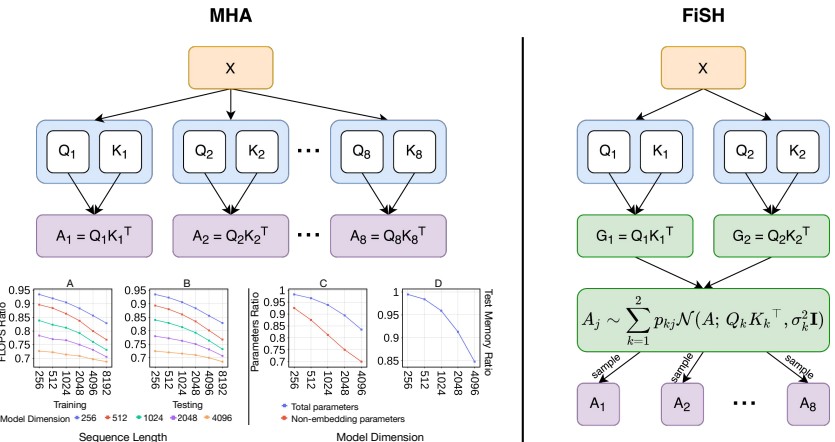

Figure 1: Our proposed finite admixture model of shared heads (FiSH) vs. the standard multi-head (MHA) attention. FiSH samples local attention matrices from a finite admixture of global attention matrices. Compared to MHA, FiSH is more efficient, saving computation and memory (See Fig. 3 and Section 4).

# 1 Introduction

Transformers have become the state-of-the-art model for solving many challenging problems in natural language processing [75, 1, 17, 80, 21, 10, 35, 61] and computer vision [19, 68, 22, 73]. Transformers learn from unlabeled data effectively and take advantage of the pre-trained models on downstream tasks that involve different data modalities with limited supervision [59, 60, 21, 82, 43]. The success of transformer is credited to the multi-head self-attention (MHA) mechanism as their fundamental building block. For each token in the sequence, self-attention in transformers aggregates information from other tokens by computing a weighted average of their feature representations with a weight proportional to a similarity score between the representations. This attention mechanism allows arbitrary input-dependent interaction between tokens in the sequence where a token can pay attention to other tokens and attain a contextual representation [6, 75, 38]. Multi-head self-attention captures multiple such contextual representations, one at each head, thereby increasing the representation capacity of the self-attention. It has been argued that the representation capacity of the attention mechanism [72] and its flexibility in capturing diverse syntactic and semantic relationships [72, 76, 15, 77, 32] account for the impressive performance of transformers in practice.

## 1.1 Background: Self-Attention

For a given input sequence $\mathbf{X} := [\boldsymbol{x}_1, \cdots, \boldsymbol{x}_N]^\top \in \mathbb{R}^{N \times D_x}$ of $N$ feature vectors, self-attention transforms $\mathbf{X}$ into the output sequence $\mathbf{H}$ in the following two steps:

**Step 1.** The input sequence $\mathbf{X}$ is projected into the query matrix $\mathbf{Q}$, the key matrix $\mathbf{K}$, and the value matrix $\mathbf{V}$ via three linear transformations
$$\mathbf{Q} = \mathbf{X}\mathbf{W}_Q^\top; \mathbf{K} = \mathbf{X}\mathbf{W}_K^\top; \mathbf{V} = \mathbf{X}\mathbf{W}_V^\top,$$
where $\mathbf{W}_Q, \mathbf{W}_K \in \mathbb{R}^{D \times D_x}$, and $\mathbf{W}_V \in \mathbb{R}^{D_v \times D_x}$ are the weight matrices. We denote $\boldsymbol{Q} := [\boldsymbol{q}_1, \cdots, \boldsymbol{q}_N]^\top, \mathbf{K} := [\boldsymbol{k}_1, \cdots, \boldsymbol{k}_N]^\top$, and $\mathbf{V} := [\boldsymbol{v}_1, \cdots, \boldsymbol{v}_N]^\top$, where the vectors $\boldsymbol{q}_i, \boldsymbol{k}_i, \boldsymbol{v}_i$ for $i = 1, \cdots, N$ are the query, key, and value vectors, respectively.

**Step 2.** The output sequence $\mathbf{H} := [\boldsymbol{h}_1, \cdots, \boldsymbol{h}_N]^\top$ is then computed as follows
$$\mathbf{H} = \text{softmax}\left(\frac{\mathbf{Q}\mathbf{K}^\top}{\sqrt{D}}\right)\mathbf{V} := \text{softmax}(\frac{\mathbf{A}}{\sqrt{D}})\mathbf{V}, \tag{1}$$
where the softmax function is applied to each row of the matrix $\mathbf{A} = (\mathbf{Q}\mathbf{K}^\top)$. This matrix $\mathbf{A} \in \mathbb{R}^{N \times N}$ and its component $a_{ij}$ for $i, j = 1, \cdots, N$ are called the attention matrix and attention scores, respectively. For each query vector $\boldsymbol{q}_i$ for $i = 1, \cdots, N$, an equivalent form of Eqn. (1) to compute the output vector $\boldsymbol{h}_i$ is given by
$$\boldsymbol{h}_i = \sum_{j=1}^N \text{softmax}\left(\boldsymbol{q}_i^\top \boldsymbol{k}_j / \sqrt{D}\right) \boldsymbol{v}_j. \tag{2}$$

The self-attention computed by Eqn. (1) and (2) is called the scaled dot-product or softmax attention. In our paper, we call a transformer that uses this attention the softmax transformer. The structure

that the attention matrix $\mathbf{A}$ learns from training determines the ability of the self-attention to capture contextual representation for each token.

**Multi-head Attention (MHA)** Each output sequence $\mathbf{H}$ forms an attention head. In MHA, multiple heads are concatenated to compute the final output. Let $H$ be the number of heads and $W^O \in \mathbb{R}^{HD \times HD}$ be the projection matrix for the output. The multi-head attention is defined as

$$\text{MultiHead}(\{\mathbf{H}\}_{i=1}^{H}) = \text{Concat}(\mathbf{H}_1, \ldots, \mathbf{H}_H)\mathbf{W}^O. \tag{3}$$

## 1.2 Eigenvalue Analysis of the Attention Matrices

The multi-head mechanism allows transformers to capture more diverse attention patterns and increase the capacity of the model. However, in many practical tasks, transformers learn redundant heads [47, 78], whose learned attention matrices lie on a low-dimensional manifold [8]. To confirm this claim, in Figure 2, we follow the eigenvalue analysis in [8] and investigate the eigenvalues of the covariance matrix of vector-

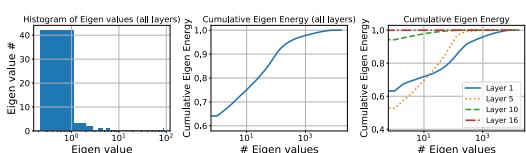

Figure 2: (Left) Histogram of the top 50 eigen values and (Middle, Right) cumulative sum of eigen values of the layer-average attention scores covariance matrix.

ized attention matrices aggregated over each layer of a transformer model trained on the WikiText-103 dataset for language modeling. We observe that this covariance matrix is low rank with top 200 (1.2%) eigenvalues capturing more than 90% of the energy. This result verifies that the variability of learned attention matrices in transformers can be explained by a relatively small number of principal components, and those attention matrices lie on a low-dimensional manifold. Therefore, in multi-head attention, the effective number of heads is much smaller than the actual number of heads, and a more effective way to compute multi-head attention is needed to improve the efficiency of transformers.

## 1.3 Contribution

Leveraging the idea of the finite admixture model (FAM) [57, 9], we propose a new class of efficient transformer architectures, namely the Transformer with a Finite Admixture of Shared Heads (FiSHformer). At the core of FiSHformer is to sample local attention matrices from an admixture of a small number of global attention matrices. This sharing mechanism between heads helps reduce the computational complexity and the model size compared to the MHA softmax transformer. Our contribution is three-fold:

1. We construct an admixture model for shared attention matrices between heads and propose FiSHformer, a novel class of transformers that take advantage of this admixture model to efficiently compute multi-head attention.

2. We introduce a nonlinearity mapping from global heads to local heads into FiSH and propose the Generalized FiSHformer (GFiSHformer). We then explore different possibilities to design FiSHformer and GFiSHformer.

3. We empirically verify that FiSHformer and GFiSHformer achieve similar or even better accuracy but with much less computational cost in terms of FLOPs and smaller model complexity measured by the number of parameters. The advantages of our methods grow with the model/feature dimension $D$ and the input sequence length $N$.

We also show that FiSHformer-based models help reduce head redundancy in our experiments.

**Organization:** We structure this paper as follows: In Section 2, we develop a finite admixture model of shared heads and then present our FiSHformer and its extensions. In Section 2.5, we analyze the reduction in model complexity and computational cost from FiSH. In Section 3 and 4, we validate and empirically analyze the efficiency and accuracy of FiSHformer, as well as conducting ablation studies on the model. We discuss related works in Section 5. The paper ends up with concluding remarks. More results and details are provided in the Appendix.

## 2 Transformer with a Finite Admixture of Shared Heads

We first review the finite admixture model (FAM) and derive a FAM of shared heads for the multi-head self-attention. We then define Transformer with a Finite Admixture of Shared Heads (FiSHformer).

## 2.1 A Probabilistic Viewpoint of Attention Matrices

Let $\mathbf{A}_j$ denote the attention matrix at the $j^{\text{th}}$ head, $j = 1, 2, \ldots, H$. From a probabilistic viewpoint, to have diversity among $\mathbf{A}_1, \ldots, \mathbf{A}_H$, we can assume that $\mathbf{A}_j$ comes from a distribution $\mathbb{P}_j$ for

all $j$. Since the distributions $\mathbb{P}_1, \ldots, \mathbb{P}_H$ can have complex forms and be difficult to compute, our approach is to consider approximated distributions of $\mathbb{P}_j$ and these approximated distributions have simple forms. A natural choice for each of these approximated distributions is via a finite mixture of Gaussian distributions, which can be summarized in the following lemma below.

**Lemma 1.** *Assume that $P \in \mathbb{R}^{D'}$ is a probability distribution supported on some compact set and admits differentiable and bounded density estimation $p$. Then, for any scale parameter $\sigma > 0$ and for any $\epsilon > 0$, there exists universal constant $C$ and $M \leq (C \log(1/\epsilon))^{D'}$ such that we can find a mixture of $M$ components $\sum_{i=1}^{M} p_i \mathcal{N}(\theta_i, \sigma^2 \mathbf{I}_{D'})$ where $p_1, \ldots, p_K$ are weight parameters and $\theta_1, \ldots, \theta_K$ are location parameters that satisfy the following inequality*

$$\sup_{x \in \mathbb{R}^d} |p(x) - \sum_{i=1}^{M} p_i \phi(x|\theta_i, \sigma^2 \mathbf{I}_{D'})| \leq \epsilon,$$

*where $\phi(.|\theta, \sigma^2 \mathbf{I})$ is Gaussian density function with location parameter $\theta$ and covariance matrix $\sigma^2 \mathbf{I}_{D'}$.*

The proof of Lemma 1 is in Appendix E. In light of Lemma 1, for each scale parameter $\sigma > 0$ and for each distribution $P_j$, we can find the corresponding number of components $M_j$, weight parameters $p_{1j}, \ldots, p_{M_j j}$, and location parameters $\theta_{1j}, \ldots, \theta_{M_j j}$ such that the mixtures $\mathbb{P}'_j = \sum_{i=1}^{M_j} p_{ij} \mathcal{N}(\theta_{ij}, \sigma^2 \mathbf{I}_{D'})$ can approximate the distribution $P_j$ up to a given accuracy $\epsilon$. However, these approximations still involve $\prod_{j=1}^{H} M_j$ number of location parameters, which can be computationally expensive. To overcome this issue, we assume that $M_1 = M_2 = \ldots = M_H$ and the location parameters $(\theta_{1j}, \ldots, \theta_{M_j j}) = (\theta_1, \ldots, \theta_M)$ for all $j$, i.e., these approximated mixtures share a similar set of location parameters. This sharing information of location parameters has a deep connection to finite admixture models, which we are going to elaborate in the next sections.

### 2.1.1 Background

Finite admixture models (FAM) are extensions of finite mixture models (FMMs), which served as a workhorse in stochastic modeling. A finite mixture distribution of $M$ components for a random array $\mathbf{X} \in \mathbb{R}^{N \times J}$ is given by

$$\boldsymbol{x}_j \sim \sum_{k=1}^{M} p_k f(\boldsymbol{x}; \theta_k), \quad \sum_{k=1}^{M} p_k = 1, \quad p_k \geq 0, \tag{4}$$

where $\boldsymbol{x}_j \in \mathbb{R}^N$ is the $j$-th row of $\mathbf{X}$ randomly sampled from the mixture distribution, $f$ is a chosen probability measure, such as a Gaussian distribution, $p = \{p_1, p_2, \ldots, p_M\}$ are mixture weights, and $\theta_k$ denotes the parameter values for the $k$-th component.

A FAM is a generalization of a FMM where rows $\boldsymbol{x}_j, j = 1, \ldots, H$, are drawn from different mixture distributions that share the components $f(\boldsymbol{x}; \theta_k), k = 1, \ldots, M$ but with different mixture weights

$$\boldsymbol{x}_j \sim \sum_{k=1}^{M} p_{kj} f(\boldsymbol{x}; \theta_k), \quad \sum_{k=1}^{M} p_{kj} = 1, \quad p_{kj} \geq 0. \tag{5}$$

## 2.2 Multi-head as a Finite Admixture Model of Shared Heads (FiSH)

As demonstrated in Section 2.1, we propose a Finite Admixture Model of Shared Heads (FiSH), in which $\mathbf{A}_j$ follows finite admixture distribution of M components given by

$$\mathbf{A}_j \sim \sum_{k=1}^{M} p_{kj} f(\mathbf{A}; \theta_k), \quad \sum_{k=1}^{M} p_{kj} = 1, \quad p_{kj} \geq 0. \tag{6}$$

Here $M < H$ and $f(\mathbf{A}; \theta_k)$ are chosen probability measures. In particular, we choose $f(\mathbf{A}; \theta_k)$ to be Gaussian distributions $\mathcal{N}(\mathbf{A}; \mathbf{G}_k, \boldsymbol{\Sigma}_k)$, where $\mathbf{G}_k = \mathbf{Q}_k \mathbf{K}_k^\top$ and $\boldsymbol{\Sigma}_k = \sigma_k^2 \mathbf{I}$ are the cluster means and covariances, respectively. FiSH is then defined as follows:

**Definition 1** (Finite Admixture Model of Shared Heads). *The multi-head attention admits a finite admixture model of shared heads if the attention matrices $\mathbf{A}_j$ at the $j^{th}$ head are sampled from the following finite admixture model:*

$$\mathbf{A}_j \sim \sum_{k=1}^{M} p_{kj} \mathcal{N}(\mathbf{A}; \mathbf{Q}_k \mathbf{K}_k^\top, \sigma_k^2 \mathbf{I}), \quad \sum_{k=1}^{M} p_{kj} = 1, \quad p_{kj} \geq 0. \tag{7}$$

In FiSH defined in Def. 1, we call $\{\mathbf{G}_k = \mathbf{Q}_k\mathbf{K}_k^\top\}_{k=1,\dots,M}$ global attention matrices and $\{\mathbf{A}_j\}_{j=1,\dots,H}$ local attention matrices. FiSH computes $M$ global attention matrices $\mathbf{G}_k$, and $H$ local attention matrices $\mathbf{A}_j$ are sampled from FiSH as in Eqn. (7) with $M < H$.

**Remark 1** (FiSH vs. Baseline MHA). *The baseline MHA with $H$ heads need to compute $H$, e.g. $H = 8$, attention matrices, each of which requires $\mathcal{O}(N^2)$ computational costs where $N$ is the length of the input sequence. In contrast, FiSH only need to compute $M < H$, e.g. $M = 2$, global attention matrices, each of which also requires $\mathcal{O}(N^2)$ computational and memory costs. Then FiSH combines those global attention matrices to form a FAM from which $H$, e.g. $H = 8$, local attention matrices are sampled as in Eqn. (7). This second step of sampling local attention matrices from a set of global attention matrices in FiSH requires very few computations. Thus, FiSH is more efficient than MHA.*

**Remark 2** (Connection to Topic Models). *FiSH can be interpreted as a Probabilistic Latent Semantic Analysis (pLSA) model for topic modeling. Considering the document $d$ that contains the word $w$ whose topic is $c$, pLSA models the occurrence of the word $w$ in the document $d$ as a mixture of conditionally independent Multinomial distributions $p(w|d) = \sum_c p(c|d)p(w|c)$. Comparing this pLSA with Eqn. (7) of FiSH, we can associate the mixture weights $p_{kj}$ and the distribution $\mathcal{N}(\mathbf{A}; \mathbf{Q}_k\mathbf{K}_k^\top, \sigma_k^2\mathbf{I})$ in FiSH with the distributions $p(c|d)$ and $p(w|c)$ in pLSA, respectively. Therefore, it can be interpreted that the global attention matrices in FiSH play the role of topics, and the local attention matrices in FiSH are words sampled from those topics. It is interesting to note that pLSA is equivalent to the famous Latent Dirichlet Allocation model under a uniform Dirichlet prior on the per-document topic distribution $p(c|d)$.*

## 2.3 Transformer with a Finite Admixture of Shared Heads

FiSHformers are transformers that use FiSH instead of MHA. FiSH, as defined in Def. 1, is not differentiable, which poses a difficulty in training FiSHformers. Applying the reparameterization trick [39], the attention matrices $\mathbf{A}_j$ can be written in a differentiable form as follows:

$$\mathbf{A}_j = \sum_{k=1}^{M} p_{kj}(\mathbf{Q}_k\mathbf{K}_k^\top + \sigma_k \odot \epsilon_j), \ \ \epsilon_j \sim \mathcal{N}(0, \mathbf{I}), \ \ \sum_{k=1}^{M} p_{kj} = 1, \ \ p_{kj} \geq 0. \tag{8}$$

FiSHformers use the formulation of local attention heads in Eqn. (8) to implement FiSH.

**Transformer with a Hard Finite Admixture of Shared Heads (Hard FiSHformer)** Hard FiSHformer takes the zero-noise limit of Eqn. (8) to reduce the computational cost. The attention matrices $\mathbf{A}_j$ in Hard FiSHformer are then calculated as

$$\mathbf{A}_j = \sum_{k=1}^{M} p_{kj}\mathbf{Q}_k\mathbf{K}_k^\top, \ \ \sum_{k=1}^{M} p_{kj} = 1, \ \ p_{kj} \geq 0. \tag{9}$$

**Remark 3** (Discriminative Relaxation). *To take the advantage of learning from data, the convex combination condition of $p_{kj}$, i.e. $\sum_{k=1}^{M} p_{kj} = 1, \ p_{kj} \geq 0$, can be relaxed, and those mixing coefficients are made learnable parameters that are learned from data during training*

**Remark 4** (Transformers with a Mixture of Shared Heads). *A transformer with a mixture of shared heads (MiSHformer) can be used to reduce the amount of computation with the cost of accuracy reduction. MiSH is a special case of FiSH when the mixture weights $p_{kj}$ are the same for all $j$. The local attention matrices $\mathbf{A}_j$ in MiSHformer are given by*

$$\mathbf{A}_j = \sum_{k=1}^{M} p_k(\mathbf{Q}_k\mathbf{K}_k^\top + \sigma_k \odot \epsilon_j), \ \ \epsilon_j \sim \mathcal{N}(0, \mathbf{I}), \ \ \sum_{k=1}^{M} p_k = 1, \ \ p_k \geq 0. \tag{10}$$

*An empirical comparison between FiSHformer and MiSHformer is provided in Section 4.*

## 2.4 Transformer with a Generalized Finite Admixture of Shared Heads

In order to increase the representation capacity of attention heads, we follow a common approach in learning representation by replacing the linear mapping in Eqns. (8) and (9) by a nonlinear mapping such as a neural network with the rectified linear units (ReLU). The Transformer with a Generalized Finite Admixture of Shared Heads (GFiSHformer) is then formulated as

$$\mathbf{A}_j = \sum_{k=1}^{M} \phi(p_{kj}(\mathbf{Q}_k\mathbf{K}_k^\top + \sigma_k \odot \epsilon_j)), \ \ \epsilon \sim \mathcal{N}(0, \mathbf{I}),$$

where $\phi$ is a nonlinear mapping and $p_{kj}$ are relaxed to be learnable parameters. Similarly, we formulate local attention matrices $\mathbf{A}_j$ in the Transformer with a Generalized Hard Finite Admixture of Shared Heads (Hard GFiSHformer) as $\mathbf{A}_j = \sum_{k=1}^{M} \phi(p_{kj}\mathbf{Q}_k\mathbf{K}_k^\top)$.

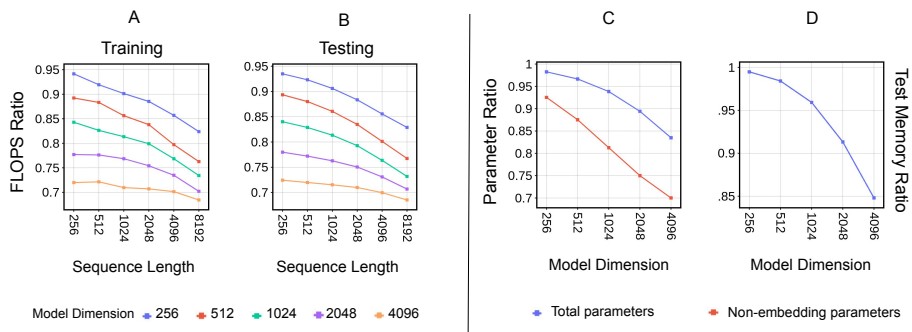

Figure 3: (Left) Training (A) and Inference (B) FLOP ratios between a 2-global-head GFiSHformers with 8-head MHA baselines across different model dimensions $D$ and sequence lengths $N$ trained on the WikiText-103 language modeling task. (Right) Number of parameters (C) and GPU memory usage at test time (D) ratios between 2-global-head GFiSHformers with 8-head MHA baselines across different model dimensions $D$. 2-global-head GFiSHformers are significantly more efficient than the baseline as $D$ and $N$ increase, indicating the benefits of our method for long-range and large-scale tasks.

Table 1: Perplexity (PPL) on WikiText-103 compared to the baselines.

| Method | Valid PPL | Test PPL |
|---|---|---|
| *Softmax 8 heads* | 33.15 | 34.29 |
| *Softmax 4 heads* | 34.80 | 35.85 |
| Hard FiSHformer 4 global heads | 33.10 | 34.11 |
| Hard FiSHformer 2 global heads | 34.14 | 35.24 |
| FiSHformer 4 global heads | 33.15 | 34.16 |
| FiSHformer 2 global heads | 34.01 | 34.96 |
| Hard GFiSHformer 4 global heads | 32.70 | 33.75 |
| Hard GFiSHformer 2 global heads | 33.31 | 34.63 |
| GFiSHformer 4 global heads | **32.68** | **33.71** |
| GFiSHformer 2 global heads | 33.21 | 34.48 |

## 2.5   Reduction in Model Complexity and Computational Cost from FiSH

Compared to its $H$-head MHA counterpart, a FiSH attention of $M$ global heads and $H$ local heads saves $[2(H-M)D - 2MH]N^2 + 2(H-M)D(2D_x - 1)N$ FLOPs in a forward pass and $2(H-M)DD_x - HM - M$ parameters. Detailed derivations are provided in Appendix D.

# 3   Experimental Results

In this section, we empirically study the advantages of FiSHformer on various tasks and benchmarks, including language modeling on WikiText-103 (Section 3.1), machine translation on IWSLT' 14 De-En and WMT'14 (Section 3.2), image classification on ImageNet (Section 3.3), time series classification on the UEA benchmark (Section 3.4), and reinforcement learning on the D4RL Benchmark (Section 3.5). We aim to show that: (i) FiSHformers improve the efficiency and accuracy upon the MHA baseline; (ii) FiSH is a universal method that can be applied to state-of-the-art transformer models to improve their performance on large-scale applications. In Section 4, we also show that FiSH helps reduce the redundancy between attention heads.

We compare FiSHformers, Hard FiSHformers, GFiSHformers, and Hard GFiSHformers with the baseline MHA softmax transformers. In our experiments, we apply the discriminative relaxation explained in Remark 3 on our FiSHformers to make the mixture weights $p_{kj}$ learnable parameters. For GFiSHformers/Hard GFiSHformers, we choose the nonlinear mapping $\phi$ to be a ReLU followed by a linear neural network. All of our results are averaged over 5 runs with different seeds. More details on datasets, models, and training are provided in Appendix A. Our PyTorch code with documentation can be found at https://github.com/minhtannguyen/FishFormer.

## 3.1   WikiText-103 Language Modeling

**Models and baselines**  We compare the 2 and 4-global-head FiSHformers with the 8-head softmax transformers [75]. Each model has 16 layers, and our training follows the setting from [66].

**Results**  Perplexity: Table 3.1 demonstrates that our 2/4-global-head (G)FiSHformers and their hard versions obtain comparable or better PPLs than the corresponding 8 head MHA baseline on

Table 2: Machine translation BLEU scores of 2-global-head (G)FiSHformers compared to the 4-head baseline on the IWSLT14 De-En dataset. Our methods perform on par or better than the baseline while being more efficient.

| Method | BLEU score |
|---|---|
| *Softmax 4 heads* | 34.42 |
| Hard Fishformer 2 global heads | 34.31 |
| FiSHformer 2 global heads | 34.38 |
| Hard GFiSHformer 2 global heads | 34.54 |
| GFiSHformer 2 global heads | **34.71** |

Table 3: Machine translation BLEU scores of 2-global-head (G)FiSH + linear transformers compared to the 4-head baseline linear on the IWSLT14 De-En benchmark. Our methods significantly outperform the linear baseline.

| Method | BLEU score |
|---|---|
| *Linear 4 heads* | 28.22 |
| Hard LFiSHformer 2 global heads | **30.24** |
| LFiSHformer 2 global heads | 29.63 |
| Hard GLFiSHformer 2 global heads | 29.20 |
| GLFiSHformer 2 global heads | 29.46 |

WikiText-103. Interestingly, the 2-global-head (G)FiSHformers perform on par with the 8-head baseline even though only 2 global attention matrices are used to span all local attention matrices, indicating that the attention matrices in MHA are indeed redundant and the representation capacity of local attention matrices in (G)FiSH, though being generated from only 2 global bases, is comparable to those in the 8-head MHA.

Efficiency: In Fig. 3A and 3B, we presents the reduction ratio of train and test FLOPS, respectively, of our 2-global-head GFiSHformer vs. the baseline 8-head MHA transformer as functions of model dimension $D$ and sequence length $N$. In Fig. 3C and 3D, we show the reduction ratio of model size and GPU memory usage at test time, respectively, of our 2-global-head GFiSHformer vs. the same baseline. We observe that the efficiency advantage of GFiSHformer over the baseline grows with $D$ and $N$, making it more suitable and superior for large-scale applications. Note that the model size in terms of the number of parameters does not depend on the sequence length $N$, and from our experiments, we observe that the GPU memory usage reduction ratio is almost the same for different sequence lengths. More efficiency analysis results on this language modeling task are provided in Section 4 and Appendix C. Also, Figure 5 in Appendix B.2 shows the train and test PPL of (G)FiSHformers and the MHA softmax transformers. In our experiment, 4-global-head GFiSHformers obtain the best validation PPL.

## 3.2 Machine Translation

In this section, we examine the performance of (G)FiSHformer on the neural machine translation task, an important task in natural language processing in which the sequence lengths of the input are not the same. We first compare (G)FiSHformers with the baselines MHA softmax transformers on the IWSLT' 14 De-En [11] and then scale up our experiments to the WMT'14 En-De [45]. On these tasks, we calculate the BLEU scores for evaluation.

**Models and baselines** For the IWSLT' 14 De-En task, we compare 2-global-heads (G)FiSHformers with the baseline 4-head softmax transformer. Each model consists of 12 layers, 6 layers for an encoder and the other 6 layers for a decoder. Our experiments follow the setting on fairseq. For the WMT'14 En-De task, we use similar models as in the IWSLT' 14 De-En task. However, we compare (G)FiSHformers of 8 and 4 global heads with the 16-head MHA softmax baseline. Our training and model setting are the same as those in [53].

**Results** As shown in Table 2 and 4, (G)FiSHformers outperform or at least are on par with the baseline MHA softmax transformers. Again, these results indicate rich representations of the local attention matrices generated by (G)FiSH. Furthermore, (G)FiSHformer outperforming Hard (G)FiSHformer in all settings suggests the positive value of adding noise into the models to turn them into a proper probabilistic model. Nevertheless, it is worth noticing that Hard (G)FiSHformer is more efficient than (G)FiSHformer. Fig. 4 in Appendix B.1 summarizes the advantage in efficiency of 2-global-head GFiSHformer over the 4-head baseline on the IWSLT' 14 De-En task. These advantages of GFiSHformer grow with the model dimension $D$.

## 3.3 Image Classification on ImageNet

The advantages of (G)FiSHformers also hold across different data modalities. To illustrate this point, in this section, we apply (G)FiSH to Swin transformer [44], a state-of-the-art vision transformer architecture, for the image classification task on the ImageNet dataset [20]. The baseline Swin-T we use has a total of 12 layers, across 4 stages of transformer blocks with 3, 6, 12, and 24 heads each. Our GFiSH Swin-T uses 6 and 12 global attention heads at the last two stages. The model and training follow the settings in [44]. We summarize our results in Table 5. Our GFiSHformer is only

Table 4: BLEU scores of (G)FiSHformers, with various numbers of shared heads compared to the 16-head baseline on the WMT'14 En-De machine translation. Our methods obtain comparable/better results than the baseline while being more efficient.

| Method | BLEU score |
|---|---|
| *Softmax 16 heads* | 29.38 |
| Hard FiSHformer 8 global heads | 29.32 |
| FiSHformer 8 global heads | **29.57** |
| Hard GFiSHformer 8 global heads | 29.27 |
| GFiSHformer 8 global heads | 29.42 |
| Hard GFiSHformer 4 global heads | 28.97 |
| GFiSHformer 4 global heads | 29.34 |

Table 5: ImageNet Image Classification accuracy scores, FLOPs, and number of parameters on Swin Transformer, comparing between baseline Swin-T and our GFiSH Swin-T. Baseline results from [44] are provided in parentheses. The Swin-T baseline uses 12 and 24 attention heads in the last two stages while our GFiSH Swin-T uses only 6 and 12 global attention heads in the last two stages.

| Method | Acc top-1 | Acc top-5 | FLOPs $(10^9)$ | Params |
|---|---|---|---|---|
| *Softmax-12/24* | **81.20** | **95.50** | 4618.24 | 28.3M |
| Hard-GFiSH-6/12 | 81.11 | 95.44 | 4372.48 | 26.2M |

Table 6: The GFiSHformer vs. the baseline softmax transformer on the UEA Time Series Classification Archive benchmark [5]. The GFiSHformer performs on par with the baseline while being more efficient. We also include the reported results from [83] and [81] (in parentheses) in addition to our reproduced results. The experiment setups and configurations for the baseline and our GFiSHformer are the same as in [81] (for the PEMS-SF, SelfRegulationSCP2, UWaveGestureLibrary datasets) and [83] (for other tasks).

| Dataset/Model | Baseline softmax | GFishformer |
|---|---|---|
| EthanolConcentration | 32.08 (33.70) | **33.70** |
| FaceDetection | **68.70 (68.10)** | 68.57 |
| HandWriting | **32.08 (30.50)** | 31.55 |
| HeartBeat | 75.77 (77.60) | **76.10** |
| JapaneseVowels | **99.46 (99.40)** | 99.37 |
| PEMS-SF | **82.66 (82.10)** | 82.66 |
| SelfRegulationSCP1 | **91.46 (92.50)** | 90.56 |
| SelfRegulationSCP2 | 54.72 (53.90) | **54.81** |
| SpokenArabicDigits | 99.33 (99.30) | **99.34** |
| UWaveGestureLibrary | 84.45 (85.60) | **85.01** |
| Average Accuracy | 72.07 (72.27) | **72.17** |

Table 7: The GFiSHFormer vs. the baseline softmax transformer on the continuous control tasks from D4RL benchmark [29]. The GFiSHFormer yields comparable results to the baseline while being more efficient. We also include the reported results from [81] (in parentheses) in addition to our reproduced results.

| Environment/Model | Baseline softmax | GFiSHFormer |
|---|---|---|
| MEDIUM-EXPERT | | |
| HalfCheetah | **91.03 (83.80)** | 90.25 |
| Hopper | 110.30 (104.40) | **110.60** |
| Walker | **108.70 (107.70)** | 108.30 |
| MEDIUM-REPLAY | | |
| Hopper | 85.61 (79.70) | **85.89** |
| MEDIUM | | |
| HalfCheetah | **42.28 (42.40)** | 41.35 |
| Hopper | 61.47 (64.20) | **63.44** |
| Walker | **68.68 (70.60)** | 67.07 |
| Avg Reward | **81.19 (79.00)** | 80.99 |

slightly more efficient than the baseline in this case because the sequence length N per window for this task is small, i.e. $N = 49$. However, like with the previous language tasks and as pointed by formula of computational cost and model complexity reduction in Section 2.5, these advantages grow with larger $D$ and $N$.

### 3.4 UEA Time Series Classification

We compare the accuracy of the GFiSHformer and the baseline softmax transformers trained on the UEA Time Series Classification Archive benchmark [5]. In Table 6, we show that GFiSHformers perform on par with the baselines. For each classification task in this benchmark, the number of GFiSHformer's global heads is half the number of heads in the baseline softmax transformers. The experiment setups and configurations for the baseline and our GFiSHformer are the same as in [81] (for the PEMS-SF, SelfRegulationSCP2, UWaveGestureLibrary datasets) and [83] (for other tasks).

### 3.5 Reinforcement Learning on the D4RL Benchmark

We also demonstrate the benefits of GFiSHformers on reinforcement learning. In Table 7, we report the results of the GFiSHformer and the softmax transformer trained for the continuous control tasks from D4RL benchmark [29] to evaluate the model performance on the offline reinforcement learning. On average, the 2-global-heads GFiSHformers perform comparably with the 4-head transformer baselines. For this benchmark, we follow the architecture and training configuration from [81].

### 3.6 FiSHformer is more effective than other methods for head-redundancy reduction

To futher demonstrate the effectiveness of our method, we compare FiSHformers against the head-redundancy reduction method in [16] on the WMT'14 machine translation task. [16] proposes the adaptively sparse transformer (AST), reducing redundancy within each head by zeroing out low-attention scores. In comparison, the results further confirm the effectiveness of our method since the

Table 8: Layer-Average mean and variance of $\mathcal{L}_2$ distances between heads of models trained for the WikiText-103 language modeling task.

| Method | Mean | Variance |
|---|---|---|
| *Softmax 8 heads* | 1.62 | 0.66 |
| *Linear 8 heads* | 1.90 | 0.06 |
| GFiSHformer 2 global heads | 2.93 | 2.62 |
| GFiSHformer 4 global heads | **3.59** | **3.95** |
| GFiSHformer 6 global heads | 3.37 | 2.78 |

Table 9: Layer-average number of principal components for 95% variance explained of the covariance of attention matrices (WikiText-103 models).

| Method | Mean |
|---|---|
| *Softmax 8 heads* | 296 |
| *Linear 8 heads* | 436 |
| GFiSHformer 2 global heads | 895 |
| GFiSHformer 4 global heads | **1408** |
| GFiSHformer 6 global heads | 1228 |

BLEU score of FiSHformer and GFiSHformer are 27.26 and 27.67, respectively, better than that of AST, which is 26.93. All models share the same architecture, with 12 transformer layers, 6 encoder, and 6 decoder layers. Our FiSHformers have 4 global heads and 8 local heads per layer while 8 attention heads are used in each AST layer.

### 3.7 Beyond Multi-Head Softmax Transformers

We show that (G)FiSH can be applied on top of many transformer architectures to improve their performance including the linear transformers [37] and the SoTA transformer with noisy back-translation [26]. More results of combining (G)FiSH with efficient transformers are in the Appendix.

**Applying (G)FiSH on Linear Transformers** Linear transformers [37] is a class of efficient transformers that linearize the softmax kernel in Eqns. 1 and 2 when computing attention matrices. We apply (G)FiSH on linear transformers trained for the IWSLT14 De-En machine translation task and summarize the results in Table 3. The empirical results verify that applying (G)FiSH using only 2-global heads on a 4-head linear transformer improves the accuracy of the baseline model.

**(G)FiSH Improves the State-of-the-Art Noisy Back-Translation** We apply an 8-global-head Hard GFiSH on the transformers trained with noisy back-translation [26] for the WMT'14 En-De translation task and obtain the BLEU score of 33.45. This result is comparable to the SoTA result of 33.52 from the transformers trained with noisy back-translation but our model is more efficient.

## 4 Empirical Analysis

We study models trained for the WikiText-103 language modeling task in this section.

**Efficiency Analysis** In this section, we further investigate the efficiency reduction of 2-global-head GFiSHformers over the 8-head baseline as a function of the number of heads in Fig. 7 and 8 and compare the efficiency of our FiSH-based models in Fig. 6. Fig. 6, 7, 8 and details on our setting are provided in Appendix C. From Fig. 7 and 8, we observe that when using fewer number of global heads, GFiSHformers achieve significantly more computation reduction (in both training and inference). Furthermore, Fig. 6 shows that the efficiency measures, i.e. FLOPs, model size, and GPU memory usage, of all FiSH-based models we study in this paper follow similar patterns.

**FiSHformer Helps Reducing Head Redundancy** We show that (G)FiSHformers attain more significant distances between heads than the baseline. Thus, our models capture more diverse patterns across heads than the baseline. For a given pre-trained model, we compute the pair-wise $\mathcal{L}_2$ distances between heads in the same layer. We show the layer-average mean and variance of distances between heads in GFiSHformers compared with those in the MHA softmax baselines in Table 8. We provide additional results for Hard GFiSHformers and Hard FiSHformer in Table 12 in the Appendix.

**Eigen Analysis** We show that heads in GFiSHformer lie on a higher-dimensional subspace compared to those in FiSHformer. This justifies our use of nonlinearity mapping to generate local heads from global heads. Using a pre-trained model, we first compute the covariance matrix of the vectorized attention scores of the $l$-th layer: $\mathcal{C}^l = \frac{1}{M \cdot H} \sum_{m=1}^{M} \sum_{h=1}^{H} (A_m^{l,h})(A_m^{l,h})^\top$. We use spectral decomposition to represent $\mathcal{C}^l$ in terms of eigenvalues and eigenvectors, namely, $\mathcal{C}^l = \sum_{i=1}^{n^2} \lambda_i v_i v_i^\top$. Without losing generality, we assume that eigenvalues are sorted in descending order. We illustrate the layer-average number of principle components that are needed to explain 95% variance in Table 9. Interestingly, Table 9 shows that attention matrices in all of our proposed FiSH-based models lie on higher-dimensional subspace than those in the baseline MHA softmax transformers, which indicates that our models achieve better representational capacity than the baseline, confirming the advantage of (G)FiSH over MHA. Table 13 in the Appendix provide additional results for Hard (G)FiSHformer.

Table 10: Perplexity (PPL) on WikiText-103 of 2-global-head FiSHformer vs. 2-global-head MiSHformer compared to the 8-head baseline. MiSHformer attains worse PPL than FiSHformer.

| Method | Valid PPL | Test PPL |
|---|---|---|
| *Softmax 8 heads* | **33.15** | **34.29** |
| FiSHformer 2 global heads | 34.01 | 34.96 |
| MiSHformer 2 global heads | 35.11 | 36.28 |

Table 11: Perplexity on WikiText-103 of GFiSHformer with various number of global heads compared with the 8-head baseline.

| Method | Valid PPL | Test PPL |
|---|---|---|
| *Softmax 8 heads* | 33.15 | 34.29 |
| GFiSHformer 6 global heads | 32.80 | 33.80 |
| GFiSHformer 4 global heads | **32.68** | **33.71** |
| GFiSHformer 2 global heads | 33.21 | 34.48 |

**Admixture vs. Mixture of Heads** We compare the transformer with a mixture of heads and the transformer with an admixture of heads. We show that the transformer with a mixture of heads yields worse accuracy. We summarize our results on the WikiText-103 language modeling task in Table 10.

**Ablation Study on the Effect of the Number of Global Heads on FiSH-based Models** We investigate the accuracy, efficiency, and representation capacity of FiSH-based models under different numbers of global heads on the WikiText-103 language modeling task. Since GFiSH obtains the best accuracy on this task, we use GFiSH as a study case in this section and report our results on accuracy in Table 11. Ablation results on efficiency are summarized in Fig. 7, 8 in Appendix C, and ablation results on representation capacity are provided in Table 8, 9.

## 5   Related Work

**Efficient Transformers** Efficient transformers have been developed to reduce the quadratic computational and memory cost of transformers [62]. A class of efficient transformers are sparse transformers which design the attention matrix to have sparse structure [55, 42, 58, 12, 7]. Another class of efficient transformers are models that integrate different access patterns for better coverage [12, 33]. These access patterns can also be learned from the data [40, 62, 71]. In other works, a side memory module is used to access multiple tokens simultaneously [41, 69, 3, 7]. Recently, low-rank and kernelization methods have been proposed to improve the computational and memory efficiency of computing self-attention [74, 79, 37, 14, 50, 67, 52, 56]. Our (G)FiSHformers are complementary to these methods.

**Redundancy in Transformers** Most of the neurons and heads in the pre-trained transformer are redundant and can be pruned when applied on downstream tasks [18, 47, 23]. The contextualized embeddings in pre-trained networks under this redundancy have also been studied to demonstrate that the representations learned within these models are highly anisotropic [48, 27]. Knowledge distillation and sparse approximation have also been used to enhance the efficiency of transformers, including [65, 70, 78, 64]. Our FiSH-based approach are complementary to these methods

**Mixture Models for Transformers** Recently, mixture models have been employed to study and enhance transformers. Among these works is switch transformers [28] that uses the routing algorithm in Mixture of Experts (MoE) to decrease the communication and computational costs in transformers. [49] derives a GMM for each attention head. Other works that combine mixture models with transformers include [51, 13, 31, 36].

## 6   Concluding Remarks

In this paper, we proposed the FiSHformer, a class of transformers that samples local attention matrices from a finite admixture of global attention matrices. FiSHformers and their generalized version GFiSHformers attain better computational cost and model complexity than their baseline MHA softmax transformer counterparts. Furthermore, (G)FiSHformers help increase the diversity between attention heads. It is worth noting that there is no potential negative societal impacts of FiSHformers. Also, global attention matrices in FiSHformers currently do not have any structure, and this is a limitation of our model. It is interesting to impose additional structures such as low-rank and sparsity into the global attention matrices. Finally, establishing theoretical guarantee for optimization algorithms [25, 24, 34] for solving FiSHformers is also an important future direction to improve the efficiency of FiSHformers.

## Acknowledgements

This material is based on research sponsored by the AFOSR MURI FA9550-18-1-0502, the ONR grant N00014-20-1-2093, the MURI N00014-20-1-2787, and the NSF under Grant# 2030859 to the Computing Research Association for the CIFellows Project (CIF2020-UCLA-38). NH acknowledges support from the NSF IFML 2019844 and the NSF AI Institute for Foundations of Machine Learning.

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
