# OpenReview forum: "Improving Transformer with an Admixture of Attention Heads"
_NeurIPS.cc/2022/Conference — NeurIPS 2022 Accept_

### Official Review · Reviewer_tCZN · 2022-07-09

**Rating:** 6
**Confidence:** 4
**Soundness:** 3 good
**Presentation:** 4 excellent
**Contribution:** 3 good

**Summary:**

This paper introduces FiSHformers, a Transformer model with a Finite Admixture of Shared Heads. FiSHformers calculate $M < H$ global heads and construct/sample the full attention heads from the $M$ heads, thus reducing a certain level of computations in constructing Transformer models.

**Questions:**

1. In this paper, the authors assume $A_j$ comes from different distributions ( the learned attention heads are diverse ) but with the same M and $\theta$. I was wondering how this assumption is justified.

2. Is it possible to measure and compare the sampled set of $A_j$s and original $A_j$s ? I mean, if there is a way to show that we can always construct original attention heads from sampled attention heads, or the former contains the latter, the claims on heads redundancy may be stronger.

3. How to determine the number of $M$ for better or similar performance.

**Limitations:**

No mentions of limitations, but do analyze the social impact.

**Strengths And Weaknesses:**

Strengths:

Overall, the idea is interesting. The analysis from a probabilistic viewpoint and applying the admixture model to attention calculation is novel. The paper is well organized and presented. The experimental results presented in this paper show that Fishformers obtained comparable or better results than Transformer models with the calculation of global attention matrices. In addition, the authors show that the purposed structure can also work with some linear Transformers.

Weaknesses:

My main concern is about the significance of the work.

1. There are no theoretical guarantees that we can obtain the same or better performance with a strictly less number of global shared components.

2. Although the number of FLOPs is shown to be less (less global attention is calculated, depends on $H$) using FiSHFormer model and its variants, the computation complexity remains the same (depends on $N$).

3. The author also mentions the head redundancy issue, but $L_2$ distances among heads are more like a reflection but not determinating factors for the claim that FiSHFormer is more efficient than other Transformer variants (low-rank, parameter sharing heads, etc) in terms of the issue.

---

> ### Author Response · Authors · 2022-08-02
> **Response to Reviewer tCZN (1)**
>
> Thank you for your thoughtful review and valuable feedback. Below we address your concerns.
>
> -----
>
> **Q1. There are no theoretical guarantees that we can obtain the same or better performance with a strictly less number of global shared components. How to determine the number of M for better or similar performance?**
>
> **Reply:**  We believe there is a misunderstanding of the motivation for using global share components in our Finite Admixture of Shared Heads (FiSH). Please allow us to clear this misunderstanding by clarifying why FiSH is a good model for the multihead attention (MHA). From our eigenvalue analysis of the attention matrices in Section 1.2, we observe that in the multihead attention, the attention matrices, i.e. $QK^{\top}$ lie on a low-dimensional manifold, and therefore,  the effective number of heads, i.e. the dimensionality of the low-dimensional manifold formed by attention matrices,  is much smaller than the actual number of heads. As a result, in FiSH, we propose to sample local attention matrices from a set of small global attention matrices. However, we do not argue that FiSH can obtain the same or better performance with a strictly less number of global attention matrices. The number of global attention matrices needs to be greater than or equal to the dimensionality of the low-dimensional manifold which the attention matrices in the MHA lie on. Note that this number is much smaller than the number of heads in the MHA. If we use fewer global attention matrices than this number, then the FiSHformer will suffer from reduction in accuracy. In our paper, we consider this number of global attention matrices as a hyper parameter to finetune. We observe that in all of our experiments, choosing the global attention matrices in FiSH to be 1/4  and 1/2 of the number attention heads in the original MHA results in models with good accuracy and efficiency. We have made these points clear at the beginning of Appendix A in our revision.
>
> **Q2. Although the number of FLOPs is shown to be less (less global attention is calculated, depends on H) using FiSHFormer model and its variants, the computation complexity remains the same (depends on N).**
>
> **Reply:** We agree with the reviewer that FiSH does not reduce the computational complexity in terms of the sequence length N. FiSH is a framework to share the attention matrices between attention heads via the set of global attention matrices, which help reduce the computational cost of the model. Furthermore, FiSH can be applied to different types of attention mechanisms. In particular, in order to reduce the computational complexity from $\mathcal{O}(N^{2})$ to $\mathcal{O}(N)$, we apply FiSH to the linear attention in [Katharopoulos et al. (2020)] . Table 3 in our paper shows that applying 2-global-head GFiSH and FiSH on the linear transformers help improve the BLEU score on the IWSLT14 De-En benchmark compared to the 4-head linear attention. Similar to FiSH and GFiSH applied to the multihead softmax attention, our linear FiSHformer and GFiSHformer are more efficient than the baseline linear transformers.
>
> **References**
>
> [1] Angelos Katharopoulos et al.. Transformers are rnns: Fast autoregressive transformers with linear attention. In ICML, pages 5156–5165. PMLR, 2020.

---

> > ### Author Response · Authors · 2022-08-02
> > **Response to Reviewer tCZN (2)**
> >
> > **Q3. The author also mentions the head redundancy issue, but L2 distances among heads are more like a reflection but not determinating factors for the claim that FiSHFormer is more efficient than other Transformer variants (low-rank, parameter sharing heads, etc) in terms of the issue.**
> >
> > **Reply:** We believe there is a misunderstanding of our results in Table 6, 7 and 3.  Please allow us to clear this misunderstanding by clarifying the key results from these tables. In Table 6 and 7, we show that GFiSHformer captures more diverse attention patterns across heads compared to the MHA softmax baseline by comparing the layer-average L2 distances between heads and the layer-average number of principal components for 95% variance explained of the covariance of attention matrices. In Table 3, we demonstrate that applying our FiSH and GFiSH on linear transformers  [Katharopoulos et al. (2020)] helps improve the performance of those linear models. To show that GFiSH is more effective than the linear attention in [Katharopoulos et al. (2020)] in reducing the head redundancy, we have also computed the layer-average L2 distances between heads and the layer-average number of principal components for 95% variance explained of the covariance of attention matrices in the linear transformer trained for the WikiText-103 language modeling task. We have included these new results in Table 6 and 7 in our revision. We observe that both results are smaller than those of the GFiSHformer trained on the same task, suggesting that the GFiSHformer learns more diverse attention patterns across heads and has less head redundancy than the linear transformer.
> >
> > **References**
> >
> > [1] Angelos Katharopoulos et al.. Transformers are rnns: Fast autoregressive transformers with linear attention. In ICML, pages 5156–5165. PMLR, 2020.
> >
> > **Q4. In this paper, the authors assume Aj comes from different distributions ( the learned attention heads are diverse) but with the same M and θ. I was wondering how this assumption is justified.**
> >
> > **Reply:** Given the result from Lemma 1 in our paper, each distribution $P_{j}$ of the attention matrix $A_{j}$ can be well approximated by a Gaussian mixture model (GMM) of $M_{j}$ components whose parameters are $\theta_{1j}, …, \theta_{M_{j}j}$. However, from our eigenvalue analysis of the attention matrices in Section 1.2, we observe that in the multihead attention, the attention matrices $A_{j}$ lie on a low-dimensional manifold. This observation motivates us to allow the GMMs for attention matrices $A_{j}$ to share components, i.e., those GMMs share the same $M$ and $\theta$. This constraint also helps reduce the computational cost, the size, and the memory usage of the model.
> >
> > **Q5. Is it possible to measure and compare the sampled set of $A_{j}$ and original $A_{j}$ ? I mean, if there is a way to show that we can always construct original attention heads from sampled attention heads, or the former contains the latter, the claims on heads redundancy may be stronger.**
> >
> > **Reply:** Thanks for your suggestion. Since the true distributions of the attention matrices $A_{j}$ is unknown and they can also be in complex forms, it is non-trivial to compare the sampled set of $A_{j}$ from FiSH and the original ones. It is indeed an interesting research direction to find what the true distribution underlying the attention matrices in transformers is and we will explore it in future work.
> >
> > -----
> >
> > We hope we have cleared your concerns about our work. We have also revised our manuscript according to your comments, and we would appreciate it if we can get your further feedback at your earliest convenience.

---

> > > ### Author Response · Authors · 2022-08-06
> > > **Response to Reviewer tCZN - Any further questions on our current draft**
> > >
> > > We would like to thank you again for your thoughtful reviews and valuable feedback.
> > >
> > > We would appreciate it if you could let us know if our responses have addressed your concerns and whether you still have any other questions on the current draft and our rebuttal.
> > >
> > > We would be happy to do any follow-up discussion or address any additional comments.

---

> > ### Comment · Reviewer_tCZN · 2022-08-07
> > **Thank you for the effort in the discussion.**
> >
> > I appreciate the effort in answering the question raised and the clarification. I'll raise the score to 6.

---

> > > ### Author Response · Authors · 2022-08-07
> > > **Thanks for your endorsement!**
> > >
> > > Thanks for your response and we appreciate your endorsement.

---

### Official Review · Reviewer_ofWV · 2022-07-10

**Rating:** 7
**Confidence:** 4
**Soundness:** 3 good
**Presentation:** 2 fair
**Contribution:** 3 good

**Summary:**

The paper proposes to use a Transformer with a Finite Admixture of  Shared Heads.
The idea is to directly learn these attention matrices rather than the local ones as in other  transformers. At the core of the method they sample local attention matrices from an 73 admixture of a small number of global attention matrices.
The paper shows numerous results on various benchmarks which support their claims.


**Questions:**

I see areas of similarity to the work: https://arxiv.org/abs/2110.08678 (Improving Transformers with Probabilistic Attention Keys). Please explain how your work differs.

**Limitations:**

Please see questions section.

**Strengths And Weaknesses:**

Strengths:
I am not familiar with the literature, but the results are well established.
The experiments show advantages of their method, pointing to new directions of improving transformers.
The paper is fairly clearly written.

Weaknesses:
The authors' description of the experimental results are not accurate enough.
I would be happy to see results on more benchmarks.

---

> ### Author Response · Authors · 2022-08-02
> **Response to Reviewer ofWV (1)**
>
> Thank you for your thoughtful review and valuable feedback. Below we address your concerns.
>
> -----
>
> **Q1. The authors' description of the experimental results are not accurate enough. I would be happy to see results on more benchmarks.**
>
> **Reply:** Thanks for your suggestion. **In order to further justify the advantage of (G)FiSHformer, we have conducted additional experiments on a set of 10 multivariate datasets from the UEA Time Series Classification Archive [Bagnall et al. (2018)] and the continuous control tasks from D4RL benchmark [Fu et al. (2020)]  to evaluate the model performance on the offline reinforcement learning**. In both experiments, we still observe that the (G)FiSHformer achieves comparable or even better performance than the MHA baseline while being much more efficient. We summarize our results below and in Appendix F of our revision.
>
> Also, please allow us to clarify the key results from our experiments. In our experiments, we show that:
>
> (1) GFiSHformer and FiSHformer with a small number of global heads  achieve similar or even better accuracy than the baseline softmax transformer with multihead attention (MHA) but **with much less computational cost** in terms of FLOPs, smaller model complexity measured by the number of parameters, and **much less GPU memory usage**. The advantages of our methods **grow with the model dimension D and the input sequence length N**.
>
> (2) When increasing the number of global heads, both GFiSHformer and FiSHformer outperform the baseline MHA significantly.
>
> (3) GFiSH and FiSH can be applied to different types of attention mechanisms, including the linear attention, and the state-of-the-art model such as the noisy back-translation.
>
> (4) The advantages of our GFiSHformer and FiSHformer models still **hold for different tasks across different data modalities** including the Wikitext-103 language modeling task, the  IWSLT14 De-En and WMT’ 14 En-De machine translation task, and the ImageNet image classification task.
>
> *On Point (1)*, Figure 3 in our paper verifies that GFiSHformer helps reduce the number of FLOPs, parameters, and GPU memory usage significantly when the model dimension D and the sequence length N increases. Note that the number of parameters and the GPU memory usage at test time of the model are independent of the sequence length. In this figure, we plot the ratio of the number of FLOPs, the number of parameters, and the test-time GPU memory usage between the 2-global-head GFiSHformer and the 8-head MHA baseline for the WikiText-103 language modeling task. Furthermore, in Figure 4 in Appendix B.3, we show that the real time (seconds/iteration), the number of parameter, and the GPU memory usage ratios between a 2-global-head GFiSHformers with 4-head MHA baselines across different model dimensions D trained on the IWSLT14 De-En machine translation task. Similarly, our GFiSHformer is more efficient than the baseline MHA, and the advantage of our model grows with the model dimension.
>
> Note that compared to the FiSHformer, the GFiSHformer applies additional nonlinear mapping to the linear combination of global attention heads to generate local attention heads. As a results, the FiSHformer has the same number of parameters as the GFiSHformer and is more efficient than the GFiSHformer in terms of FLOPs and test-time memory usage. A mathematical analysis on the efficiency advantage of (G)FiSHformer vs. the MHA baseline is summarized in Section 2.5 in the main text with detailed derivation provided in Appendix D.
>
> On accuracy, in Table 1, we shows that the GFiSHformer with 2 global heads obtains comparable results with the 8-head softmax MHA baseline (34.48 test PPL vs. 34.29 test PPL) and outperforms the 4-head softmax MHA baseline (34.48 test PPL vs. 35.85 test PPL). Furthermore, Table 2 in our paper demonstrates that the 2-global-head GFiSHformer attains better BLUE score than the 4-head MHA baseline while  the 2-global-head FiSHformer obtains comparable BLUE score as the baseline on the IWSLT14 De-En machine translation task. Similarly, Table 4 shows that our 8-global-head GFiSHformer and FiSHformer achieve better BLEU score than the 16-head MHA baseline on the WMT’ 14 En-De machine translation task. Finally, Table 5 demonstrates the potential of applying GFiSHformer to vision transformers, showing that GFiSHformer with 6 and 12 global attention heads in the last two stages attains similar top-1 and top-5 accuracy on the ImageNet image classification task as the MHA baseline with 12 and 24 attention heads in the last two stages.

---

> > ### Author Response · Authors · 2022-08-02
> > **Response to Reviewer ofWV (2)**
> >
> > *On Point (2)*, Table 1 in our paper shows that both GFiSHformer and FiSHformer with 4 global heads outperform the 8-head MHA baseline on the WikiText-103 language modeling task (33.71 and 34.11 test  PPL vs. 34.29 test PPL, respectively). Furthermore, in Figure 7 and 8 in Appendix C, we show that our GFiSHformer with 4 global heads still achieves much better efficiency than the 8-head MHA baseline in terms of FLOPs, the number of parameters, and GPU memory usage at test time.
> >
> > *On Point (3)*, Table 3 in our paper shows that applying 2-global-head GFiSH and FiSH on the linear transformers [Katharopoulos et al. (2020)] help improve the BLEU score on the IWSLT14 De-En benchmark compared to the 4-head linear attention. In Section 3.5 of our paper, we also show that an 8-global-head GFiSHformer trained with noisy back-translation for the WMT’14 En-De translation task obtain the BLEU score of 33.45, which is comparable to the state-of-the-art result of 33.52 reported in [Edunov et al. (2018)]. In both cases, our GFiSHformer and FiSHformer are more efficient than the baseline.

---

> > > ### Author Response · Authors · 2022-08-02
> > > **Response to Reviewer ofWV (3)**
> > >
> > > *On Point (4)*, as we have described above, experiments in our paper show the advantage of (G)FiSHformer over the MHA baseline on different tasks that involve different data modality (i.e. text and images) including the Wikitext-103 language modeling task, the  IWSLT14 De-En and WMT’ 14 En-De machine translation task, and the ImageNet image classification task. Our additional experiments, as mentioned at the beginning of this reply, show that (G)FiSHformer achieves comparable or even better performance than the MHA baseline while being much more efficient on UEA Time Series Classification Archive benchmark [Bagnall et al. (2018)] and the D4RL benchmark for offline reinforcement learning [Fu et al. (2020)]. We summarize our results below and in Appendix F of our revision.
> > >
> > > In our paper, we also compute the layer-average distance between attention heads to show that our GFiSHformer and FiSHformer capture more diverse attention patterns and reduce the head redundancy (see Table 6 and 7 in the main text and Table 10 and 11in Appendix B). This can serve as a justification for the advantage of GFiSHformer and FiSHformer in terms of accuracy.
> > >
> > > Table 1: The GFiSHformer vs. the baseline softmax transformer on the UEA Time Series Classification Archive benchmark. We also include the reported results from [Zerveas et al. (2021)] and [Wu et al. (2022)] (in parentheses) in addition to our reproduced results.
> > >
> > > | Dataset/Model       | Baseline softmax          | GFishformer   |
> > > | :---        |    :----:   |    :----:   |
> > > | EthanolConcentration     |   32.08 (33.70)  |     **33.70**    |
> > > | FaceDetection     |    **68.70** (68.10)   |     68.57    |
> > > | HandWriting    |    **32.08** (30.50)    |      31.55    |
> > > | HeartBeat    |    75.77 (77.60)    |      **76.10**    |
> > > | JapaneseVowels    |    **99.46** (99.40)  |     99.37   |
> > > | PEMS-SF    |    **82.66** (82.10)    |      **82.66**  |
> > > | SelfRegulationSCP1    |    **91.46** (92.50)  |     90.56  |
> > > | SelfRegulationSCP2   |    54.72 (53.90)   |      **54.81**  |
> > > | SpokenArabicDigits   |    99.33 (99.30)    |     **99.34** |
> > > | UWaveGestureLibrary   |    84.45 (85.60)    |    **85.01** |
> > > | **Average Accuracy**   |   72.07 (72.27)    |     **72.17** |
> > >
> > >
> > > Table 2: The GFiSHFormer vs. the baseline softmax transformer on the continuous control tasks from D4RL
> > > benchmark. We also include the reported results
> > > from [Wu et al. (2022)]  (in parentheses) in addition to our reproduced results.
> > >
> > > | Environment/Model       | Baseline softmax          | GFishformer   |
> > > | :---        |    :----:   |    :----:   |
> > > |         |    Medium-Expert   |     |
> > > | HalfCheetah     |   **91.03** (83.80)  |     90.25   |
> > > | Hopper     |    110.30 (104.40)   |     **110.60**    |
> > > | Walker   |    **108.70** (107.70)    |      108.30    |
> > > |         |    Medium-Replay   |     |
> > > | Hopper     |   **85.61** (79.70)  |     85.89   |
> > > |         |    Medium   |     |
> > > | HalfCheetah     |   **42.28** (42.40)  |     41.35   |
> > > | Hopper     |    61.47 (64.20)   |     **63.44**    |
> > > | Walker   |    **68.68** (70.60)    |      67.07    |
> > > | **Avg Reward**   |   **81.19** (79.00)    |     **80.99** |
> > >
> > > **References**
> > >
> > > [1] Angelos Katharopoulos et al.. Transformers are rnns: Fast autoregressive transformers with linear attention. In ICML, pages 5156–5165. PMLR, 2020.
> > >
> > > [2] Sergey Edunov et al. Understanding back-translation at scale. In Proceedings of the 2018 Conference on Empirical Methods in Natural Language Processing, pages 489–500. ACL.
> > >
> > > [3] Anthony Bagnall et al. The uea multivariate time series classification archive. arXiv preprint arXiv:1811.00075, 2018.
> > >
> > > [4] Justin Fu et al. D4rl: Datasets for deep data-driven reinforcement learning. arXiv preprint arXiv:2004.07219, 2020.
> > >
> > > [5] George Zerveas et al. A transformer-based framework for multivariate time series representation learning. In Proceedings of the 27th ACM SIGKDD Conference on Knowledge Discovery & Data Mining,pages 2114–2124, 2021.
> > >
> > > [6] Haixu Wu et al. Flowformer: Linearizing transformers with conservation flows. In ICML, 2022.

---

> > > > ### Author Response · Authors · 2022-08-02
> > > > **Response to Reviewer ofWV (4)**
> > > >
> > > > **Q2. I see areas of similarity to the work: https://arxiv.org/abs/2110.08678 (Improving Transformers with Probabilistic Attention Keys). Please explain how your work differs.**
> > > >
> > > > **Reply:** There are three main differences between our FiSH model and the Probabilistic Attention Keys in [Nguyen et al. (2022)]. We explain these differences below and have discussed [Nguyen et al. (2022)] in the Related Work section of our revision.
> > > >
> > > > (1) First, [Nguyen et al. (2022)] models each attention head as a Gaussian mixture model (GMM) and shows that the posterior distribution from this GMM matches the attention score. In their GMM, each Gaussian component is centered around an attention key, and the attention queries are sampled from this GMM.  In contrast, our FiSH models multihead attention as a finite admixture of Gaussians, where each global attention head (i.e. global attention matrix) follows a Gaussian distribution. We sample local attention heads from our finite admixture of Gaussians model.
> > > >
> > > > (2) Second, the Gaussian mixture model in [Nguyen et al. (2022)] is a finite mixture model. In their model, the attention queries are sampled from the same mixture distribution. The finite admixture model is a generalization of a finite mixture model in which samples are drawn from different mixtures distributions (see Section 2.1.1 in our main text). In the case of FiSH, the local attention heads are sampled from different mixtures of Gaussians that share the same Gaussian components, each of which is centered at a global attention head, but with different mixture weights (see Section 2.1 and 2.2 in our main text).
> > > >
> > > > (3) Finally, [Nguyen et al. (2022)] extends their GMM models into a Mixture of Gaussian Keys in which they replace each Gaussian component in their GMM by another GMM. This is a mixture of mixture model, in which multiple attention keys are used at each token position. [Nguyen et al. (2022)] argues that by using more attention keys at each token position, they can reduce the number of heads in multihead attention without reduction in accuracy. On the contrary, in our paper, we argue that attention heads lie on a low-dimensional manifold (see Figure 2 in our paper and [Bhojanapalli et al. 2021]). We capture this property of attention heads in the multihead attention by a finite admixture model and develop FiSH. FiSH samples local attention heads from an admixture of a small number of global attention heads.
> > > >
> > > > Despite their differences, FiSH and the Mixture of Gaussian Keys in [Nguyen et al. (2022)] are complementary. In particular, we can use the Mixture of Gaussian Keys in [Nguyen et al. (2022)] to generate the global attention heads and then use FiSH to sample the local attention heads from this set of global attention heads.
> > > >
> > > > **References**
> > > >
> > > > [1] Tam Minh Nguyen et al.. Improving transformers with probabilistic attention keys. In ICML, pages 16595–16621. PMLR,2022.
> > > >
> > > > [2] Srinadh Bhojanapalli et al.. Eigen analysis of self-attention and its reconstruction from partial computation. arXiv preprint arXiv:2106.08823, 2021.
> > > >
> > > > -----
> > > >
> > > > We hope we have cleared your concerns about our work. We have also revised our manuscript according to your comments, and we would appreciate it if we can get your further feedback at your earliest convenience.

---

> > > > > ### Author Response · Authors · 2022-08-06
> > > > > **Response to Reviewer ofWV - Any further questions on our current draft**
> > > > >
> > > > > We would like to thank you again for your thoughtful reviews and valuable feedback.
> > > > >
> > > > > We would appreciate it if you could let us know if our responses have addressed your concerns and whether you still have any other questions on the current draft and our rebuttal.
> > > > >
> > > > > We would be happy to do any follow-up discussion or address any additional comments.

---

### Official Review · Reviewer_eA9U · 2022-07-14

**Rating:** 7
**Confidence:** 3
**Soundness:** 3 good
**Presentation:** 3 good
**Contribution:** 4 excellent

**Summary:**

The authors report the transformer model with global attention heads rather than the local ones as in other types of transformer architectures.

**Questions:**

It would be interesting to see if the admixture of attention heads is applicable for the image domain as feature sharing is more natural there. Have the authors tried that?

**Ethics Review Area:**

["I don’t know"]

**Limitations:**

Yes

**Strengths And Weaknesses:**

Strengths:

The paper is well-written and interesting to read, math is described sufficiently. Learning shared information over the transformer head can be a well scalable feature that can be used in other transformer-like architectures.

Weaknesses:

The authors performed several benchmarks, however, the improvement over the vanilla model is marginal. Even though the experiment was performed for 5 runs with various random seeds, it is hard to judge on this improvement as various parameters can affect performance.

---

> ### Author Response · Authors · 2022-08-02
> **Response to Reviewer eA9U (1)**
>
> Thank you for your thoughtful review and valuable feedback. Below we address your concerns.
>
> -----
>
> **Q1. The authors performed several benchmarks, however, the improvement over the vanilla model is marginal. Even though the experiment was performed for 5 runs with various random seeds, it is hard to judge on this improvement as various parameters can affect performance.**
>
> **Reply:** We believe there is a misunderstanding of the advantage of our FiSHformer over the baselines. Please allow us to clear this misunderstanding by clarifying the key results from our experiments. In our experiments, we show that:
>
> (1) GFiSHformer and FiSHformer with a small number of global heads  achieve similar or even better accuracy than the baseline softmax transformer with multihead attention (MHA) but **with much less computational cost** in terms of FLOPs, smaller model complexity measured by the number of parameters, and **much less GPU memory usage**. The advantages of our methods **grow with the model dimension D and the input sequence length N**.
>
> (2) When increasing the number of global heads, both GFiSHformer and FiSHformer outperform the baseline MHA significantly.
>
> (3) GFiSH and FiSH can be applied to different types of attention mechanisms, including the linear attention, and the state-of-the-art model such as the noisy back-translation.
>
> (4) The advantages of our GFiSHformer and FiSHformer models still **hold for different tasks across different data modalities** including the Wikitext-103 language modeling task, the  IWSLT14 De-En and WMT’ 14 En-De machine translation task, and the ImageNet image classification task.
>
> *On Point (1)*, Figure 3 in our paper verifies that GFiSHformer helps reduce the number of FLOPs, parameters, and GPU memory usage significantly when the model dimension D and the sequence length N increases. Note that the number of parameters and the GPU memory usage at test time of the model are independent of the sequence length. In this figure, we plot the ratio of the number of FLOPs, the number of parameters, and the test-time GPU memory usage between the 2-global-head GFiSHformer and the 8-head MHA baseline for the WikiText-103 language modeling task. Furthermore, in Figure 4 in Appendix B.3, we show that the real time (seconds/iteration), the number of parameter, and the GPU memory usage ratios between a 2-global-head GFiSHformers with 4-head MHA baselines across different model dimensions D trained on the IWSLT14 De-En machine translation task. Similarly, our GFiSHformer is more efficient than the baseline MHA, and the advantage of our model grows with the model dimension.
>
> Note that compared to the FiSHformer, the GFiSHformer applies additional nonlinear mapping to the linear combination of global attention heads to generate local attention heads. As a results, the FiSHformer has the same number of parameters as the GFiSHformer and is more efficient than the GFiSHformer in terms of FLOPs and test-time memory usage. A mathematical analysis on the efficiency advantage of (G)FiSHformer vs. the MHA baseline is summarized in Section 2.5 in the main text with detailed derivation provided in Appendix D.
>
> On accuracy, in Table 1, we shows that the GFiSHformer with 2 global heads obtains comparable results with the 8-head softmax MHA baseline (34.48 test PPL vs. 34.29 test PPL) and outperforms the 4-head softmax MHA baseline (34.48 test PPL vs. 35.85 test PPL). Furthermore, Table 2 in our paper demonstrates that the 2-global-head GFiSHformer attains better BLUE score than the 4-head MHA baseline while  the 2-global-head FiSHformer obtains comparable BLUE score as the baseline on the IWSLT14 De-En machine translation task. Similarly, Table 4 shows that our 8-global-head GFiSHformer and FiSHformer achieve better BLEU score than the 16-head MHA baseline on the WMT’ 14 En-De machine translation task. Finally, Table 5 demonstrates the potential of applying GFiSHformer to vision transformers, showing that GFiSHformer with 6 and 12 global attention heads in the last two stages attains similar top-1 and top-5 accuracy on the ImageNet image classification task as the MHA baseline with 12 and 24 attention heads in the last two stages.
>
> *On Point (2)*, Table 1 in our paper shows that both GFiSHformer and FiSHformer with 4 global heads outperform the 8-head MHA baseline on the WikiText-103 language modeling task (33.71 and 34.11 test  PPL vs. 34.29 test PPL, respectively). Furthermore, in Figure 7 and 8 in Appendix C, we show that our GFiSHformer with 4 global heads still achieves much better efficiency than the 8-head MHA baseline in terms of FLOPs, the number of parameters, and GPU memory usage at test time.

---

> > ### Author Response · Authors · 2022-08-02
> > **Response to Reviewer eA9U (2)**
> >
> > *On Point (3)*, Table 3 in our paper shows that applying 2-global-head GFiSH and FiSH on the linear transformers [Katharopoulos et al. (2020)] help improve the BLEU score on the IWSLT14 De-En benchmark compared to the 4-head linear attention. In Section 3.5 of our paper, we also show that an 8-global-head GFiSHformer trained with noisy back-translation for the WMT’14 En-De translation task obtain the BLEU score of 33.45, which is comparable to the state-of-the-art result of 33.52 reported in [Edunov et al. (2018)]. In both cases, our GFiSHformer and FiSHformer are more efficient than the baseline.
> >
> > *On Point (4)*, as we have described above, experiments in our paper show the advantage of (G)FiSHformer over the MHA baseline on different tasks that involve different data modality (i.e. text and images) including the Wikitext-103 language modeling task, the IWSLT14 De-En and WMT’ 14 En-De machine translation task, and the ImageNet image classification task. In order to further justify the advantage of (G)FiSHformer, we have conducted additional experiments on a set of 10 multivariate datasets from the UEA Time Series Classification Archive [Bagnall et al. (2018)] and the continuous control tasks from D4RL benchmark [Fu et al. (2020)] to evaluate the model performance on the offline reinforcement learning. In both experiments, we still observe that the (G)FiSHformer achieves comparable or even better performance than the MHA baseline while being much more efficient. We summarize our results below and in Appendix F of our revision.
> >
> > In our paper, we also compute the layer-average distance between attention heads to show that our GFiSHformer and FiSHformer capture more diverse attention patterns and reduce the head redundancy (see Table 6 and 7 in the main text and Table 10 and 11in Appendix B). This can serve as a justification for the advantage of GFiSHformer and FiSHformer in terms of accuracy.
> >
> > Table 1: The GFiSHformer vs. the baseline softmax transformer on the UEA Time Series Classification Archive benchmark. We also include the reported results from [Zerveas et al. (2021)] and [Wu et al. (2022)] (in parentheses) in addition to our reproduced results.
> >
> > | Dataset/Model       | Baseline softmax          | GFishformer   |
> > | :---        |    :----:   |    :----:   |
> > | EthanolConcentration     |   32.08 (33.70)  |     **33.70**    |
> > | FaceDetection     |    **68.70** (68.10)   |     68.57    |
> > | HandWriting    |    **32.08** (30.50)    |      31.55    |
> > | HeartBeat    |    75.77 (77.60)    |      **76.10**    |
> > | JapaneseVowels    |    **99.46** (99.40)  |     99.37   |
> > | PEMS-SF    |    **82.66** (82.10)    |      **82.66**  |
> > | SelfRegulationSCP1    |    **91.46** (92.50)  |     90.56  |
> > | SelfRegulationSCP2   |    54.72 (53.90)   |      **54.81**  |
> > | SpokenArabicDigits   |    99.33 (99.30)    |     **99.34** |
> > | UWaveGestureLibrary   |    84.45 (85.60)    |    **85.01** |
> > | **Average Accuracy**   |   72.07 (72.27)    |     **72.17** |
> >
> >
> > Table 2: The GFiSHFormer vs. the baseline softmax transformer on the continuous control tasks from D4RL
> > benchmark. We also include the reported results
> > from [Wu et al. (2022)]  (in parentheses) in addition to our reproduced results.
> >
> > | Environment/Model       | Baseline softmax          | GFishformer   |
> > | :---        |    :----:   |    :----:   |
> > |         |    Medium-Expert   |     |
> > | HalfCheetah     |   **91.03** (83.80)  |     90.25   |
> > | Hopper     |    110.30 (104.40)   |     **110.60**    |
> > | Walker   |    **108.70** (107.70)    |      108.30    |
> > |         |    Medium-Replay   |     |
> > | Hopper     |   **85.61** (79.70)  |     85.89   |
> > |         |    Medium   |     |
> > | HalfCheetah     |   **42.28** (42.40)  |     41.35   |
> > | Hopper     |    61.47 (64.20)   |     **63.44**    |
> > | Walker   |    **68.68** (70.60)    |      67.07    |
> > | **Avg Reward**   |   **81.19** (79.00)    |     **80.99** |
> >
> > **References**
> >
> > [1] Angelos Katharopoulos et al.. Transformers are rnns: Fast autoregressive transformers with linear attention. In ICML, pages 5156–5165. PMLR, 2020.
> >
> > [2] Sergey Edunov et al. Understanding back-translation at scale. In Proceedings of the 2018 Conference on Empirical Methods in Natural Language Processing, pages 489–500. ACL.
> >
> > [3] Anthony Bagnall et al. The uea multivariate time series classification archive. arXiv preprint arXiv:1811.00075, 2018.
> >
> > [4] Justin Fu et al. D4rl: Datasets for deep data-driven reinforcement learning. arXiv preprint arXiv:2004.07219, 2020.
> >
> > [5] George Zerveas et al. A transformer-based framework for multivariate time series representation learning. In Proceedings of the 27th ACM SIGKDD Conference on Knowledge Discovery & Data Mining,pages 2114–2124, 2021.
> >
> > [6] Haixu Wu et al. Flowformer: Linearizing transformers with conservation flows. In ICML, 2022.

---

> > > ### Author Response · Authors · 2022-08-02
> > > **Response to Reviewer eA9U (3)**
> > >
> > > **Q2. It would be interesting to see if the admixture of attention heads is applicable for the image domain as feature sharing is more natural there. Have the authors tried that?**
> > >
> > > **Reply:** Table 5 and Section 3.3 in our paper demonstrates the potential of applying GFiSHformer to vision transformers, showing that GFiSHformer with 6 and 12 global attention heads in the last two stages attains similar top-1 and top-5 accuracy on the ImageNet image classification task as the MHA baseline with 12 and 24 attention heads in the last two stages.
> > >
> > > -----
> > > We hope we have cleared your concerns about our work. We have also revised our manuscript according to your comments, and we would appreciate it if we can get your further feedback at your earliest convenience.

---

> > > > ### Author Response · Authors · 2022-08-06
> > > > **Response to Reviewer eA9U - Any further questions on our current draft**
> > > >
> > > > We would like to thank you again for your thoughtful reviews and valuable feedback.
> > > >
> > > > We would appreciate it if you could let us know if our responses have addressed your concerns and whether you still have any other questions on the current draft and our rebuttal.
> > > >
> > > > We would be happy to do any follow-up discussion or address any additional comments.

---

> > > > > ### Comment · Reviewer_eA9U · 2022-08-08
> > > > > **Detailed and comprehensive review, a few more remarks**
> > > > >
> > > > > Thank you for the detailed comment. As I mentioned above, I like the idea and I think it is an interesting paper. Just a few more suggestions to the authors:
> > > > > - avoid using abstract comparisons, such as "much smaller", "much less computational cost", or "much fewer floating-point operations", as a reader I would like to see the numbers.
> > > > > - Figure 3. is a bit hard to read, I would suggest putting units (sequence length) at the bottom, not "Training/Testing"
> > > > > - I am still not convinced, that the model (considerably) outperforms the vanilla transformer, since in Figure 3. the authors compare 2 global heads and 8 softmax multi-head attention (MHA) in order to show better performance, however, if we compare 8 MHA and 2 global heads (when available), we'll see that in Table 1, Table 8, and Table 9 -  8 MHA outperform 2 global heads. In other tables, the authors compare a different number of heads, but usually, MHA is double of global heads, not x4. Once the performance comparison for more global heads is not shown it is hard to judge whether the model performs with "much less computational cost".

---

> > > > > > ### Author Response · Authors · 2022-08-08
> > > > > > **Thanks for your endorsement!**
> > > > > >
> > > > > > Thanks for your further feedback and we appreciate your endorsement. We have fixed Figure 3 as you suggested in our revision.
> > > > > >
> > > > > > Regarding your concern that “it is hard to judge whether the model performs with "much less computational cost"”, **Figure 7 and 8 in Appendix C  of our paper plot the ratio of the number of FLOPs, the number of parameters, and the test-time GPU memory usage between the 2/4/6-global-head GFiSHformer and the 8-head MHA baseline for the WikiText-103 language modeling task reported in Table 1**. As can be seen in these figures, the GFiSHformer with 4 or 6 global heads is still more efficient than the 8-head MHA baseline, and this advantage of our model grows with the sequence length and the model size. In addition, as shown in Figure 4 in Appendix B.1 of our paper, on the IWSLT’14 De-En task, our 2-global-head GFiSHformer is remarkably more efficient than the 4-head MHA baseline.
> > > > > >
> > > > > > Furthermore, in Table 1 and 9 in our paper, the test PPL gap between the GFiSHformer with 2 global heads and the the 8-head MHA baseline is very small (34.48 vs. 34.29) while as can be seen in Figure 3, our GFiSHformer with 2 global heads gain much better efficiency than the 8-head MHA baseline in terms of FLOPs, the number of parameters, and the test-time GPU memory usage. Table 8 in our paper compares FiSHformer (Transformer with a Finite Admixture of Shared Heads) and MiSHformer (Transformers with a Mixture of Shared Heads, please see Remark 4 in the paper). FiSHformer in most cases yields worse accuracy than its generalized version, the GFiSHformer.
> > > > > >
> > > > > > We would also like to point out the results on the WMT’14 En-De machine translation task in Table 4 in our paper. This table shows that the GFiSHformer with only 4 global heads achieves comparable BLEU score to the 16-head MHA baseline (29.34 vs. 29.38).

---

### Author Response · Authors · 2022-08-03
**General Response (1)**

Dear AC and reviewers,

Thanks for your thoughtful reviews and valuable comments, which have helped us improve the paper significantly. We are encouraged by the endorsements that: 1) Learning shared information over the transformer head can be a well scalable feature (Reviewer eA9U); 2) The analysis from a probabilistic viewpoint and applying the admixture model to attention calculation are novel (Reviewer tCZN); and 3) Our idea is interesting (Reviewer tCZN), and experiments show advantages of the method, pointing to new directions of improving transformers (Reviewer ofWV). We have updated our submission based on the reviewers' feedback, and we have highlighted our revision in blue.

One of the common comments is that there is not sufficient empirical evidence showing the advantage of our proposed (G)FiSHformer, and results on more benchmarks are needed. We first address this comment here. Our (G)FiSHformer model has four main advantages:

(1) GFiSHformer and FiSHformer with a small number of global heads  achieve similar or even better accuracy than the baseline softmax transformer with multihead attention (MHA) but **with much less computational cost** in terms of FLOPs, smaller model complexity measured by the number of parameters, and **much less GPU memory usage**. The advantages of our methods **grow with the model dimension D and the input sequence length N**.

(2) When increasing the number of global heads, both GFiSHformer and FiSHformer outperform the baseline MHA significantly.

(3) GFiSH and FiSH can be applied to different types of attention mechanisms, including the linear attention, and the state-of-the-art model such as the noisy back-translation.

(4) The advantages of our GFiSHformer and FiSHformer models still **hold for different tasks across different data modalities** including the Wikitext-103 language modeling task, the  IWSLT14 De-En and WMT’ 14 En-De machine translation task, and the ImageNet image classification task.

*On Point (1)*, Figure 3 in our paper verifies that GFiSHformer helps reduce the number of FLOPs, parameters, and GPU memory usage significantly when the model dimension D and the sequence length N increases. Note that the number of parameters and the GPU memory usage at test time of the model are independent of the sequence length. In this figure, we plot the ratio of the number of FLOPs, the number of parameters, and the test-time GPU memory usage between the 2-global-head GFiSHformer and the 8-head MHA baseline for the WikiText-103 language modeling task. Furthermore, in Figure 4 in Appendix B.3, we show that the real time (seconds/iteration), the number of parameter, and the GPU memory usage ratios between a 2-global-head GFiSHformers with 4-head MHA baselines across different model dimensions D trained on the IWSLT14 De-En machine translation task. Similarly, our GFiSHformer is more efficient than the baseline MHA, and the advantage of our model grows with the model dimension.

Note that compared to the FiSHformer, the GFiSHformer applies additional nonlinear mapping to the linear combination of global attention heads to generate local attention heads. As a results, the FiSHformer has the same number of parameters as the GFiSHformer and is more efficient than the GFiSHformer in terms of FLOPs and test-time memory usage. A mathematical analysis on the efficiency advantage of (G)FiSHformer vs. the MHA baseline is summarized in Section 2.5 in the main text with detailed derivation provided in Appendix D.

On accuracy, in Table 1, we shows that the GFiSHformer with 2 global heads obtains comparable results with the 8-head softmax MHA baseline (34.48 test PPL vs. 34.29 test PPL) and outperforms the 4-head softmax MHA baseline (34.48 test PPL vs. 35.85 test PPL). Furthermore, Table 2 in our paper demonstrates that the 2-global-head GFiSHformer attains better BLUE score than the 4-head MHA baseline while  the 2-global-head FiSHformer obtains comparable BLUE score as the baseline on the IWSLT14 De-En machine translation task. Similarly, Table 4 shows that our 8-global-head GFiSHformer and FiSHformer achieve better BLEU score than the 16-head MHA baseline on the WMT’ 14 En-De machine translation task. Finally, Table 5 demonstrates the potential of applying GFiSHformer to vision transformers, showing that GFiSHformer with 6 and 12 global attention heads in the last two stages attains similar top-1 and top-5 accuracy on the ImageNet image classification task as the MHA baseline with 12 and 24 attention heads in the last two stages.

---

> ### Author Response · Authors · 2022-08-03
> **General Response (2)**
>
> *On Point (2)*, Table 1 in our paper shows that both GFiSHformer and FiSHformer with 4 global heads outperform the 8-head MHA baseline on the WikiText-103 language modeling task (33.71 and 34.11 test  PPL vs. 34.29 test PPL, respectively). Furthermore, in Figure 7 and 8 in Appendix C, we show that our GFiSHformer with 4 global heads still achieves much better efficiency than the 8-head MHA baseline in terms of FLOPs, the number of parameters, and GPU memory usage at test time.
>
> *On Point (3)*, Table 3 in our paper shows that applying 2-global-head GFiSH and FiSH on the linear transformers [Katharopoulos et al. (2020)] help improve the BLEU score on the IWSLT14 De-En benchmark compared to the 4-head linear attention. In Section 3.5 of our paper, we also show that an 8-global-head GFiSHformer trained with noisy back-translation for the WMT’14 En-De translation task obtain the BLEU score of 33.45, which is comparable to the state-of-the-art result of 33.52 reported in [Edunov et al. (2018)]. In both cases, our GFiSHformer and FiSHformer are more efficient than the baseline.

---

> > ### Author Response · Authors · 2022-08-03
> > **General Response (3)**
> >
> > *On Point (4)*, as we have described above, experiments in our paper show the advantage of (G)FiSHformer over the MHA baseline on different tasks that involve different data modalities (i.e. text and images) including the Wikitext-103 language modeling task, the IWSLT14 De-En and WMT’ 14 En-De machine translation task, and the ImageNet image classification task. In order to further justify the advantage of (G)FiSHformer, we have conducted additional experiments on a set of 10 multivariate datasets from the UEA Time Series Classification Archive [Bagnall et al. (2018)] and the continuous control tasks from D4RL benchmark [Fu et al. (2020)] to evaluate the model performance on the offline reinforcement learning. In both experiments, we still observe that the (G)FiSHformer achieves comparable or even better performance than the MHA baseline while being much more efficient. We summarize our results below and in Appendix F of our revision.
> >
> > In our paper, we also compute the layer-average distance between attention heads to show that our GFiSHformer and FiSHformer capture more diverse attention patterns and reduce the head redundancy (see Table 6 and 7 in the main text and Table 10 and 11in Appendix B). This can serve as a justification for the advantage of GFiSHformer and FiSHformer in terms of accuracy.
> >
> > Table 1: The GFiSHformer vs. the baseline softmax transformer on the UEA Time Series Classification Archive benchmark. We also include the reported results from [Zerveas et al. (2021)] and [Wu et al. (2022)] (in parentheses) in addition to our reproduced results.
> >
> > | Dataset/Model       | Baseline softmax          | GFishformer   |
> > | :---        |    :----:   |    :----:   |
> > | EthanolConcentration     |   32.08 (33.70)  |     **33.70**    |
> > | FaceDetection     |    **68.70** (68.10)   |     68.57    |
> > | HandWriting    |    **32.08** (30.50)    |      31.55    |
> > | HeartBeat    |    75.77 (77.60)    |      **76.10**    |
> > | JapaneseVowels    |    **99.46** (99.40)  |     99.37   |
> > | PEMS-SF    |    **82.66** (82.10)    |      **82.66**  |
> > | SelfRegulationSCP1    |    **91.46** (92.50)  |     90.56  |
> > | SelfRegulationSCP2   |    54.72 (53.90)   |      **54.81**  |
> > | SpokenArabicDigits   |    99.33 (99.30)    |     **99.34** |
> > | UWaveGestureLibrary   |    84.45 (85.60)    |    **85.01** |
> > | **Average Accuracy**   |   72.07 (72.27)    |     **72.17** |
> >
> >
> > Table 2: The GFiSHFormer vs. the baseline softmax transformer on the continuous control tasks from D4RL
> > benchmark. We also include the reported results
> > from [Wu et al. (2022)]  (in parentheses) in addition to our reproduced results.
> >
> > | Environment/Model       | Baseline softmax          | GFishformer   |
> > | :---        |    :----:   |    :----:   |
> > |         |    Medium-Expert   |     |
> > | HalfCheetah     |   **91.03** (83.80)  |     90.25   |
> > | Hopper     |    110.30 (104.40)   |     **110.60**    |
> > | Walker   |    **108.70** (107.70)    |      108.30    |
> > |         |    Medium-Replay   |     |
> > | Hopper     |   **85.61** (79.70)  |     85.89   |
> > |         |    Medium   |     |
> > | HalfCheetah     |   **42.28** (42.40)  |     41.35   |
> > | Hopper     |    61.47 (64.20)   |     **63.44**    |
> > | Walker   |    **68.68** (70.60)    |      67.07    |
> > | **Avg Reward**   |   **81.19** (79.00)    |     **80.99** |
> >
> > **References**
> >
> > [1] Angelos Katharopoulos et al.. Transformers are rnns: Fast autoregressive transformers with linear attention. In ICML, pages 5156–5165. PMLR, 2020.
> >
> > [2] Sergey Edunov et al. Understanding back-translation at scale. In Proceedings of the 2018 Conference on Empirical Methods in Natural Language Processing, pages 489–500. ACL.
> >
> > [3] Anthony Bagnall et al. The uea multivariate time series classification archive. arXiv preprint arXiv:1811.00075, 2018.
> >
> > [4] Justin Fu et al. D4rl: Datasets for deep data-driven reinforcement learning. arXiv preprint arXiv:2004.07219, 2020.
> >
> > [5] George Zerveas et al. A transformer-based framework for multivariate time series representation learning. In Proceedings of the 27th ACM SIGKDD Conference on Knowledge Discovery & Data Mining,pages 2114–2124, 2021.
> >
> > [6] Haixu Wu et al. Flowformer: Linearizing transformers with conservation flows. In ICML, 2022.
> >
> > -----
> >
> > We are glad to answer any further questions you have on our submission.

---

### Author Response · Authors · 2022-08-03
**Summary of the Revision**

Incorporating the comments and suggestions from all reviewers, besides fixing typos and notations, we have made the following main changes in the revised paper.

1. We have added additional experiments on a set of 10 multivariate datasets from the UEA Time Series Classification Archive [Bagnall et al. (2018)] and the continuous control tasks from D4RL benchmark [Fu et al. (2020)]  to evaluate the model performance on the offline reinforcement learning. Those results are discussed in Appendix F of our revision.

2. We have discussed how to decide the number M of global attention matrices at the beginning of Appendix A of our revision.

3. We have also computed the layer-average L2 distances between heads and the layer-average number of principal components for 95% variance explained of the covariance of attention matrices in the linear transformer [Katharopoulos et al. (2020)] trained for the WikiText-103 language modeling task. In Table 6 and 7 in the main text, we compare these results with those of GFiSH to show that GFiSH is more effective than the linear attention in reducing the head redundancy.

**References**

[1] Anthony Bagnall et al. The uea multivariate time series classification archive. arXiv preprint arXiv:1811.00075, 2018.

[2] Justin Fu et al. D4rl: Datasets for deep data-driven reinforcement learning. arXiv preprint arXiv:2004.07219, 2020.

[3] Angelos Katharopoulos et al.. Transformers are rnns: Fast autoregressive transformers with linear attention. In ICML, pages 5156–5165. PMLR, 2020.

---

### Author Response · Authors · 2022-08-06
**Any Questions from the Reviewers?**

Dear reviewers,

We would like to thank all reviewers again for your thoughtful reviews and valuable feedback. We would appreciate it if you could let us know if there are additional questions or concerns about our revision and rebuttal.

We would be happy to do any follow-up discussion or address any additional comments.

---

### Meta-Review · Area_Chair_B5LB · 2022-08-31

**Recommendation:** Accept
**Confidence:** Less certain

**Metareview:**

This work proposes a version of transformers with an admixture of attention heads. The reviewers find the idea interesting. They find the paper to be well organized and presented, and with sufficient empirical support for the main conclusions. There was some initial concern over similarity to prior work, but the reviewer indicated this has been resolved in the discussion with the authors. I therefore recommend accepting the paper.

**Award:**

No

---

### Decision · Program_Chairs · 2022-09-14

Accept